# Motivation toward Physical Activity and Healthy Habits of Adolescents: A Systematic Review

**DOI:** 10.3390/children10040659

**Published:** 2023-03-30

**Authors:** Javier Cachón-Zagalaz, Hanrriette Carrasco-Venturelli, María Sánchez-Zafra, María Luisa Zagalaz-Sánchez

**Affiliations:** 1Musical, Plastic and Corporal Expression Didactics Department, University of Jaén, 23071 Jaén, Spain; 2Department of Physical Activity Sciences, University of Los Lagos, Osorno 5290000, Chile

**Keywords:** motivation, adolescence, physical activity, healthy habits

## Abstract

Adolescence is a transformative period in which rapid physical, cognitive and psychosocial growth takes place. Laying the foundation for healthy behaviors is paramount during these formative years. The aim of this review is to determine which countries are leading in research on adolescents’ motivation towards physical activity and healthy habits and their main findings. A systematic review was conducted following the PRISMA statement using the Web of Science and Scopus databases during the months of September to December 2022. The search terms used were: “Physical activity”, “Motivation” and “Adolescents”, in the following research areas: Education, Educational Research and Sport Sciences. A total of 5594 articles were identified, but only those that met the established criteria were included (32 articles). It is concluded that most of the research is led in Spain with 16 articles, followed by 3 in Chile, 2 in Portugal, 2 in Norway and the rest of the countries with 1. Likewise, most of the works include very similar aspects regarding the incidence of motivation towards the adherence to the practice of Physical Activity and healthy habits.

## 1. Introduction

Adolescence is a period in which young people face the difficult task of developing their personality and identity, as well as establishing their relational network [1]. It is the stage of development located between childhood and adulthood, in which a growing process of physical, psychological and social maturation takes place, leading the human being to become an adult. Very important rapid changes occur, with which the individual becomes biologically, psychologically and socially mature and capable of trying to live independently [2]. These changes affect the quality of life, both in terms of healthy habits and the practice of Physical Activity (PA), in which diet and nutrition play an important role in maintaining health and preventing diseases. These are times when adolescents should have tools that allow them to make a good decision on health and nutrition habits [3]. Added to that is the importance of PA that is associated with improved health outcomes and reduced cognitive impairment; many of these benefits are seen with only 60 min of moderate-to-vigorous intensity of PA per day [4].

Unfortunately, as indicated by the World Health Organization (WHO) [5], 1 in 4 adults and 3 in 4 adolescents (aged 11 to 17 years) worldwide are obese. As the economic development of countries increases, inactivity increases. There are countries where levels of physical inactivity can reach 70%, which added to the fact that adolescents are more prone to unhealthy behaviors due to globalization; unhealthy nutritional behaviors have been adopted, such as nutrient-poor food, insufficient intake of all food groups, high sugar content, fast food and little PA [3], as well as convenience foods. According to the WHO [5], new threats to the health of children and adolescents, as major health risks, include the rapid increase in childhood obesity, of which has become one of the most serious public health problem of the 21st century. At present, few countries offer the necessary conditions to help all children to grow up and have a healthy future [4]. Given this scenario, life satisfaction is a key indicator of adolescent health that refers to how people examine their lives in general and in specific domains such as family, friends and leisure time [6,7].

The Physical Education (PE) class plays a decisive role in the development of students, not only in the physical area but also in the cognitive, affective, attitudinal and social areas, i.e., it contributes to the formation of the whole person [8]. The PE class should seek strategies that facilitate capturing the interest and awakening the motivation of students when carrying out their teaching activities, with the aim for this motivation to transcend the classroom, integrating into the students’ lifestyle in a healthy and replicable way in their daily life, with family, friends and in leisure spaces [9]. The main purpose of the PE class is to provide opportunities for all students to acquire the knowledge, skills and attitudes that will enable them to improve, through regular PA, their quality of life and that of others [10]. The study by Zueck et al. [11] on satisfaction with PE classes and intentionality to be active, in elementary school children, aimed to identify the impact on the degree of satisfaction and intention in the practice of PE, finding higher levels of satisfaction in the students who received the class properly planned, highlighting that among the factors that can influence motivation are the quality of the class and its planning. From this perspective, in the stage of adolescence, which is a very heterogeneous period of great physical and psychological transformation [12], the influence of different external agents, which are key elements in the motivational orientation of young people, should be taken into account [13].

However, as indicated by Säfvenbom et al. [14], the practice of PE increases by 40% from junior high to high school and more than 50% of women in junior high school do not like PE or feel that the classes should be provided in a different way. Similar results are observed by Fuentes and Lagos [15], in which the main motivations for the practice of physical sports activity (PSA) are personal capacity for sport, a means to pass the time, self-improvement and health reasons; they highlight low scores regarding non-motivation in their adolescents. In turn, in a study [16] on motivation in Chilean schoolchildren, one of the results in relation to the demotivation or non-motivation dimension showed that when faced with statements such as “I participate in the PE class, but I do not really know why” or “I do not understand why we should have PE”, the majority of respondents were neutral (indifferent). Likewise, high preferences were observed in somewhat agree and somewhat disagree. On the other hand, ref. [14] gender differences play a prominent role in the attitude towards PE, and positive behavior towards PE tends to decrease with age. Students with a more self-determined profile presented a higher level of perceived autonomy support, higher levels of intrinsic motivation and a healthier lifestyle, there being a positive relationship between student perceived autonomy support, their level of self-determined motivation towards sport and a healthy lifestyle [17].

Likewise, emotional stimuli interact to some extent with cognitive processes that affect reasoning ability, memory, attitude, disposition and interest, all of which are linked to motivation and intention [18]. The results offer insight into the potential role of multilevel factors in promoting PA, suggesting that public health intervention strategies to promote PA in adolescents should recognize multiple levels of influences that may enhance or impede the likelihood of PA among adolescents [19]. A synergy between PE classes and PA is noted: if the student experiences an increase in motivation towards PE classes, the necessary motivation could be generated to encourage their commitment to be physically active in and out of school [20].

As a result of these reviews, the following questions arise: how can motivation promoted by PE teachers, a young person’s family, context and their personal characteristics affect and influence the attitude of adolescents towards the practice of PA and healthy habits? In view of the above, it is very necessary to investigate with greater precision and different variables in order to have more background on the incidence of motivation towards physical practice and healthy habits in adolescents, given the importance of acquiring good behaviors in this line for their development throughout their lives.

The aim of this study is to determine which countries lead the main research findings on motivation towards PA and healthy habits in adolescents.

## 2. Materials and Methods

The method used for this study was the elaboration of a systematic review. For this purpose, the PRISMA 2020 statement [21] was followed in order to achieve an adequate and organized structure of the manuscript. It is a qualitative review that presents the evidence in “descriptive” form and without statistical analysis (systematic review without meta-analysis). Quantitative reviews can also present the evidence in descriptive form, but using statistical techniques to “numerically” combine the results against a point estimator called “meta-analysis” [22].

### 2.1. Search Procedure and Strategy

The search was carried out during the last weeks of September to December 2022 in the “Web of Science (WOS)” and “Scopus (SJR)” databases. The main topic of this research revolves around the impact on the motivation of adolescents to practice PA and healthy habits. The search terms used were: “Physical activity”, “Motivation” and “Adolescents”. These terms have been selected as they are considered to be the most appropriate to focus the search on research that responds to the central theme.

The search initially produced a result of 5594 papers. A time range was delimited, selecting only papers published between 2015 and 2022; a total of 3274 studies remained. The search was refined by selecting those works that were only “articles”, obtaining a count of 3099. This number was reduced to 3031 when we selected all the articles that were written in English or Spanish, and finally, 1319 studies were selected after applying the criteria of the following research areas: “Sport science”, “Education educational research”.

### 2.2. Inclusion and Exclusion Criteria

The following inclusion and exclusion criteria were established: (1) research that was only applied in the school system; (2) the sample only consisted of adolescents between 10 and 19 years of age, belonging to an educational center; (3) non-experimental cross-sectional research; (4) application of validated instruments with complementary items.

For the application of these criteria, an initial preliminary reading of the title and abstract of each article was carried out, which made it possible to discard the works that did not meet these criteria. Subsequently, a more exhaustive reading of the selected articles was carried out, leaving a final sample of 32 scientific papers, where 22 of them were in English writing journals and 11 in Spanish (Figure 1).

## 3. Results

Table 1 shows a summary of some significant data of the selected sample. The data collected include the following aspects: year of publication, authors, country, objective, conclusion, recommendations and limitations. In some of the studies, not all items are specifically detailed, and therefore, some specific data are not reflected in the table, as well as the titles of the works and journals in which they have been published and other information found in the bibliographic references.

## 4. Discussion

The aim of this study was to determine which countries are the leaders in research and their main findings on motivation for PA and healthy habits among adolescents. In the studies analyzed in the systematization of the two highest level databases, WoS and SJR, it was found that Spain leads the research conducted in this area, with 50% of the same in 16 articles. Analyzed by continent, this research is led by Europe with a total of 24 publications, equivalent to 75%, and is distributed as follows: A total of 16 in Spain, 2 in Finland, 2 in Portugal, 1 in Norway, 1 in Estonia, 1 in Belgium and 1 Lithuania. The second continent is America with six articles, corresponding to 19% of the total published: a total of three in Chile, one in the United States, one in Mexico and one in Brazil; finally, in Asia, there are two articles, representing 6% of the total published: one in China and one in Pakistan.

In the case of Spain, the research is carried out in different Autonomous Communities and their universities. Selected by the institution to which the first author belongs are: In the Autonomous Community of Andalusia, three articles: one at the University of Granada [28] and three at the University of Almeria [18,30,35]. In the Community of Castilla-León, there are two articles at the Universidad Pontificia de Salamanca [31,42]. In the Community of Catalonia, there is one article at the University of Lleida [41]. In the Community of Extremadura, there are four articles in the University of Extremadura [23,25,29,47]. In the Community of Madrid, there is one article in the Universidad Antonio de Nebrija [32]. In the Community of Murcia, there is one article in the University of Murcia [17]. In the Community of Valencia, there are two articles at the University of Valencia [37,39] and one article at the “Miguel Hernández” University of Elche [27].

In the case of the second country, Chile, two of the publications are led by Chilean authors and carried out in the Araucanía region: one article in the Catholic University of Temuco [15] and one article in the University of Playa Ancha (Valparaíso, Chile) [36]. The third article is led by the University of Jaén (Jaén, Spain), followed by other authors from the Universidad of Los Lagos (Osorno, Chile) [26].

It is relevant to highlight that in three of the articles of the systematic review, researchers who did not correspond to the country from which the sample was studied appeared as the main authors, as was the case of Hagger et al. [24], which, with the first researcher from an Australian university, developed the work in the context of students from Finland. This was additionally the case for Mayorga-Vega et al. [26], with Spanish and Chilean researchers carrying out the aforementioned study in Chile, and for the research developed with Pakistani schoolchildren, led by authors from a Chinese university working with others from Pakistan (Kiyani et al.) [44].

In relation to the objective, the collection was analyzed and the main findings of the incidences in the motivation of adolescents towards PA and healthy habits were synthesized. The articles revised mostly coincide with the influence of motivational factors in the adherence to PA and in the life satisfaction of adolescents, particularly the motivation of their self-determination to perform PA. Similarly, students with a higher level of regular PA practice registered more intrinsic motivation and identified regulation towards PE and lower motivation in those who do not practice DFA. Other personal factors, such as sport self-concept, physical self-concept or perceived competence, are also key predictors of future PA [51,52,53]. It is also observed how factors related to self-perception and those that were most frequently associated with enjoyment and potential health benefits become motivational factors that influence self-determination to perform PA among adolescents. Likewise, several results show that male adolescents have a higher level of intrinsic motivation and PA than girls. Finally, intrinsic motivation means that a person’s actions are consistent with self-approved reasons for the action (e.g., for pleasure, fun or personal interest). Men present higher values of PA practice than women in terms of self-determined demotivation toward physical exercise [54,55].

Regarding psychological predictors of PA, adolescent boys reported higher levels of self-determined motivation towards physical exercise and intention to be physically active than girls. In addition, boys show greater ego-orientation in PA practice [56,57,58]; in turn, females are more inclined to less sporty activities, whereas males tend to be more predisposed to competition [57,59,60]. However, in the systematic investigation by Hopkins et al. [53], the importance of adolescent girls’ attitudes or behavioral beliefs is highlighted when considering their motivation to participate in or continue with sports over time.

Additionally, research results show that PE has an impact on motivation towards adherence to PA practice, showing that generating strategies towards teacher autonomy support positively predicted students’ enjoyment, confidence and motivation, while a controlling teacher style negatively affected confidence, enjoyment and motivation, being associated with students’ overall intrinsic motivation, which would impact the predictability of the context of the PE class towards the adoption of healthy lifestyle habits such as proper eating and regular PA practice [61].

PE is an important subject that could help improve participation in general PA during the day; provide the possibility to develop physical exercise; improve self-concept and well-being based on eudaimonia; and acquire the knowledge, skills, motivation and habits to be active outside school hours and in later life [62]. PE is the only opportunity for some students to learn and practice sports, gymnastics, games, dance, exercise and healthy living habits [63,64]. According to the self-determination theory, the quality of motivation is a mediator that relates to behavioral, emotional and cognitive outcomes of EF [65,66,67,68]. EF teachers play a major role in the experiences of students’ need for autonomy [69]. Additionally, teacher autonomy support is associated with higher students’ autonomous motivation for leisure-time PA [70,71] and the perceived controlling behavior of EF teachers is associated with lower levels of intention to be physically active and lower out-of-school PA [72].

Regarding the second objective: to determine the main recommendations made, it can be identified in terms of the PE class that teachers should mainly consider fostering their students’ satisfaction with the practice of PA and/or sports with strategies that favor the motivational climate which is focused on perceived task involvement. They should focus on strategies that incorporate elements such as personal development, progression and effort [73], which is positively associated with a number of adaptive outcomes such as perceived competence and enjoyment [74,75], given that a student’s perceived ego-involving climate has been found to be related to maladaptive outcomes such as boredom, anxiety and feelings of decreasing enjoyment [76].

Among the proposals presented in the 32 articles, there are some coincidences, for example, that the PE class (from the educators and researchers perspective) must know the context; incorporating a variety of variables such as gender, sociometric features and anthropometrics, among others, is useful in order to make adaptations in the teaching–learning process—knowing the students and the context in which they grow and develop will allow the teacher to design learning experiences that are adapted, relevant and challenging, and that promote the progressive advance in regard to each student and his/her previous knowledge [77,78,79,80,81].

Regarding the last objective of identifying the main limitations found and/or suggested by the authors in their research that could help to guide future studies in this area of research, the aspect that they point out most as a limitation is that their investigations were cross-sectional where cause–effect cannot be seen. It is worth mentioning that within the inclusion and exclusion criteria, one condition they had to meet was that they were of this type of methodology, although longitudinal research “compares data obtained at different times or moments from the same population, with the purpose of evaluating changes” [82]. However, cross-sectional research provides information on the object of study (population or sample) only once at a given time. This limitation could be analyzed in future systematic review studies that incorporate longitudinal studies in this area.

Likewise, research using aspects such as gender, context and family aspects, among others, is recommended in order to obtain a more complete view, although the greater the number of variables to be observed in a study, the more complex it is to carry out. In addition, a more representative sample could be used, since the sample studied is a subgroup of the population from which data are collected and should be representative of that population [83]. However, the sampling method is used to estimate the size of a sample depending on the type of research to be conducted, the objectives and/or hypotheses, and the defined research design [82].

Finally, another aspect mentioned in some research is the type of questionnaire applied, some of which can be interpreted differently by researchers. In certain topics, respondents answer according to their knowledge of them; in others, they may underestimate or overestimate their answers, generating erroneous information for the researcher, affecting the reliability, validity and objectivity [83]. It is also recommended that students should be supported in their participation, given that they could respond without taking into account the objectives of the study. Undoubtedly, analyzing the limitations of this or other types of research is very necessary and advisable to help improve future research.

## 5. Conclusions

It is concluded that Spain leads the publications on this subject with 16 articles analyzed, followed by Chile with 3, and Portugal and Norway with 2 articles each.

Teachers of PE should motivate adolescents towards physical practice through methodological intervention strategies that increase their satisfaction with learning and increase their commitment to PA and healthy habits in the long term.

Most research agrees that motivation is a relevant factor in terms of the importance of adherence to the practice of PA, and that in the PE class, satisfaction with physical sports practice should be encouraged in adolescents by means of methodological strategies that favor a motivational climate.

## 6. Limitations of the Study and Future Prospects

Since these were cross-sectional studies, it has not been possible to investigate more deeply into cause–effect situations or temporal characteristics.

It is recommended that future research should carry out a more in-depth analysis with the inclusion of demographic data, combining different variables to obtain information on the cause–effect of adherence to PA practice and healthy habits in adolescents.

## Figures and Tables

**Figure 1 children-10-00659-f001:**
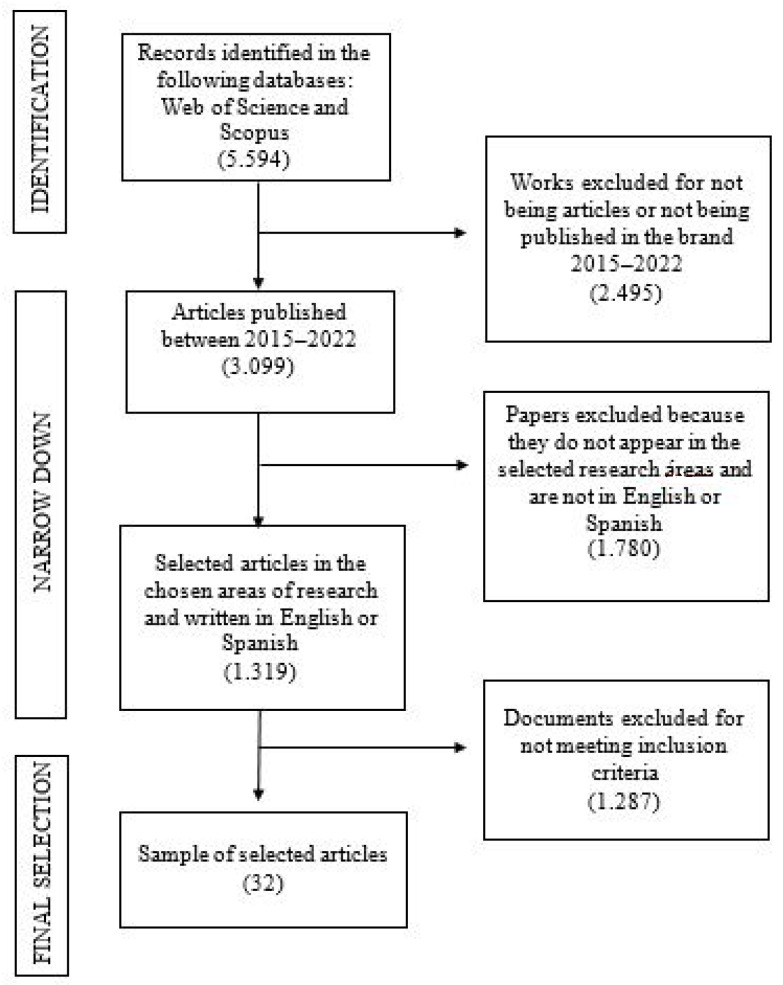
Flow chart.

**Table 1 children-10-00659-t001:** Base body of the study.

Year/Country	Authors/University	Aim	Results	Recommendations	Limitations
1. 2015 Norway	Säfvenbom et al. U. Agder [14]	To obtain data on attitudes toward and self-determined motivation for PE among a representative sample of adolescents.	PE in Norway seems to favor students who participate in competitive youth sports.	PF is experienced in contexts, which means that it, and thus development through the subject, cannot be studied in isolation. More studies on students’ FE experiences are needed and it is suggested that they be based on this perspective.	1. Longitudinal or intervention studies need to be conducted to test these hypotheses. In addition, further study is needed on how students interpret PE in terms of concepts such as ability, mastery and competence and what they identify as the dominant discourses in EF.
2. 2016 Spain	Samperio et al. U. Extremadura [23]	To analyze the relationships between motivational variables and barriers to physical exercise.	Structural equation modeling showed that perceived motivational climate involving parental homework positively predicted incremental belief. More self-determined motivation negatively predicted barriers to physical exercise.	To promote the basic psychological need for competence in order to reduce the number of barriers perceived by the subjects when performing physical exercise, and consequently avoid the abandonment of physical sports practice by adolescents.	1. In a correlational study, causal relationships cannot be established, although it provides an explanatory model. 2. Due to the problem of equivalent models presented by the structural equation technique, it is assumed that the model proposed in this study is only one of the possible models. 3. Being only one questionnaire, the data do not provide as much information as if we had carried out more.
3. 2016 Finland	Hagger et al. U. Curtin/U. Helsinki [24]	To examine the interrelationships between the quality of motivation, self-regulation techniques and PA behavior and to explore the possible mediating role of self-regulation techniques in the relationship between autonomous motivation and PA behavior.	Youth who were autonomously motivated were more likely to engage in strategic efforts to pursue those behaviors, such as planning and monitoring their progress, compared to youth who exercise for controlled reasons.	The development of interventions that support autonomous motivation for PA may encourage greater engagement in self-regulation techniques and positively affect PA behavior.	1. Reliance on self-report measures and correlational design, which limits causal inference.
4. 2017 Spain	Sánchez-Miguel et al. U. Extremadura [25]	To test the possible relationships between variables related to PA levels, as well as differences in gender and educational level, motivation, self-identity, body dissatisfaction and daily sitting time in a sample of Spanish high school adolescents.	Gender differences in the variables evaluated. Male participants showed higher intrinsic motivation and lower demotivation than females. In addition, males showed higher levels of PA than females.	The importance of promoting the intrinsic reasons for PA to encourage positive consequences in high school students.	1. The cross-sectional design of the research, which precludes making causal inferences about the impact of body size perceptions on psychological health. 2. The support of teachers and parents could be tested to improve knowledge of the factors that promote PA.
5. 2018 Spain	Mayorga-Vega et al. U. Jaén/U. Los Lagos [26]	To compare the levels of physical fitness, PA, sedentary behavior and psychological predictors of PA among male and female Chilean adolescents.	Adolescent boys have more health-promoting levels of physical fitness, habitual PA and psychological predictors of PA than girls.	The importance of promoting the intrinsic reasons for PA to encourage positive consequences in high school students.	1. The cross-sectional design does not allow causal inferences to be drawn about the relationships between the variables studied. 2. The relatively small sample. 3. Accelerometry: several authors have recognized the high rate of noncompliance as one of the most important methodological problems.
6. 2018 Spain	Moreno-Murcia et al. U. Miguel Hernández (Elche, Alicante) [27]	To examine the effects of teacher behavioral control on secondary PE students’ global intrinsic motivation, perceptions of subject matter importance, intentions to be physically active, PA level and life satisfaction.	A controlling teaching style was negatively associated with students’ global intrinsic motivation. In turn, global intrinsic motivation predicted perceived importance of the subject, which explained PA intentions that were positively associated with the level of PA, which explained life satisfaction.	Knowing the context for designing more adaptive learning environments for students.	1. Cross-sectional research cannot infer causality from correlational data. 2. Similar studies looking at different educational levels, including the use of self-observation and the inclusion of additional variables, such as gender.
7. 2018 Spain	Zurita et al. U. Granada [28]	To evaluate selected psychometric properties of the Perceived Motivational Climate in Sport Questionnaire and to study the relationship between motivational climate, PA and gender in adolescents.	Structural equation modeling identified a negative relationship between ego and task climate. This relationship was strong among women.	Teachers who create ego orientations in school could generate negative effects for students. It is important to promote task-related motivations to develop the pleasure of learning and encourage long-term PA practice.	1. Limited by its descriptive cross-sectional design, which precludes drawing conclusions about causality or directionality. 2. Differentiation of the two basic dimensions of motivational climate—task climate and ego climate—without considering the three factors that constitute each.
8. 2019 Spain	Trigueros-Ramos et al. U. Almería [18]	To analyze the relationship between the role of the teacher in relation to the structural dimensions of the teaching environment of PE and the basic psychological and self-motivation needs of adolescents as determinants of their behaviors related to eating habits and the practice of PA.	The study shows the influence and importance of the PE teacher and the motivational and emotional processes present in adolescents during PE classes on the adoption of active lifestyle habits.	It is essential to create a motivational and interesting atmosphere in the PE classes so that students can extrapolate the positive experiences that take place during the classes and take them to the sports context.	1. Since this is a correlational study, it is not possible to extrapolate cause–effect relationships. 2. The results obtained could be interpreted in many different ways depending on how the individual understands them. 3. It is recommended to use longitudinal stages that show the evolution of the relationship between the teacher and the student.
9. 2019 Spain	Vaquero et al. U. Extremadura [29]	To find out the relationship between satisfaction and frustration of basic psychological needs, motivation levels, PA and life satisfaction.	The results show the importance of motivational processes in PA and the effects of PA on life satisfaction in adolescents who devote more time to PA.	Autonomous motivation of adolescents should be fostered through autonomy support by offering alternative activities that have rational meaning.	1. According to the theory of self determination, the cross-sectional nature of the study is considered a limitation; it has shown that this theoretical model can vary according to the behavior of each group of students. Therefore, it can also be influenced by the teacher and by other agents of the affective environment, such as family and friends, and the socioeconomic environment. 2. The non-inclusion of an objective instrument for the evaluation of PA levels.
10. 2019 Spain	Trigueros et al. U. Almería [30]	To investigate the relationships between teacher’s international style of psychological needs (satisfaction and frustration), mind-wandering, mindfulness, positive and negative emotions, autonomous motivation for PE and PA classes, and intention to be physically active, in high school students.	Students who feel more autonomous see their psychological needs being met and feel emotionally positive; this will result in the development of autonomous motivation towards PE and PA classes which, in turn, could lead to a greater intention to be physically active.	Academic environments should work on and enhance positive emotions to promote student motivation and learning.	1. Regarding the findings of the model, it should be emphasized that this is a correlational study and does not allow extrapolation of cause–effect relationships. 2. The results could be interpreted in multiple ways depending on how each person understands them. 3. Future studies should analyze the influence of the social context (i.e., parents, friends) on student participation in EF classes.
11. 2019 Spain	Moral-García et al. U. Pontificia de Salamanca [31]	To analyze the relationship between the motivations generated by the PE teacher and the gender and level of PA practice of the adolescents.	Correlational analysis revealed that intrinsic motivation variables are the most related. Males were more motivated than females, except for external introductory motivation and motivation. More active adolescents were more motivated than sedentary adolescents.	It was shown that the instrument analyzed is valid and reliable. It was recommended that the seven-factor structure should be respected. The positive correlation between high levels of motivation and active boys and adolescents was observed.	1. The sample of this research could be enlarged or could be longitudinal, which would provide more causal relationships. 2. Some questions could be misinterpreted by students.
12. 2019 Spain	Trigueros et al. U. Antonio de Nebrija. Madrid [32]	To investigate how emotions (intelligence and emotional state) can influence adolescent resilience and motivation, as well as academic performance and the adoption of healthy lifestyle habits.	It demonstrates the influence and importance of PE teachers and the motivational and emotional processes of adolescents during PE classes and the role they play in the acquisition of active lifestyle habits.	Future studies should analyze in depth the results obtained using intervention studies that better clarify the relationship between the different study variables. In addition, it would also be useful to determine the influence of motivation and shame, given the variability of how they are perceived as adolescents grow older and progressively make their own independent decisions.	1. As this is a correlational study, it does not allow the extrapolation of cause–effect relationships. 2. The results could be interpreted in multiple ways depending on how each person understands them. 3. Future studies should analyze the influence of the social context (i.e., parents, friends) on student participation in PE classes.
13. 2019 Mexico	Ornelas et al. U. Autónoma de Chihuahua [33]	To compare the motivational profiles of Mexican high school students towards PE classes.	Male adolescents, in relation to female adolescents, show greater motivation towards the PE class, favoring the practice of PA to a greater extent.	The differences found between the populations studied in terms of their motivation towards the PE class suggest that when designing any type of intervention aimed at increasing this motivation, the gender variable should be taken into account.	1. Only high school students, which poses a threat to the generalizability of these results (broadening the sample). 2. Stems from the measurement instrument itself, which is based on self-report and therefore may contain biases arising from social desirability.
14. 2019 Portugal	Cid et al. Lusófona University of Humanities and Technologies [34]	To observe the motivational climate created by the teacher in the classroom, students’ satisfaction with Basic Psychological Needs (BPN) and how their behavioral regulation could explain PE grades and intention to practice sports in the future.	A learning-oriented climate has a positive impact on students’ BPN/NPB satisfaction. However, only competence satisfaction had a significant positive relationship with students’ autonomous motivation, which in turn had a significant positive relationship with EF grade as well as sport/activity intentions.	PE classes could play a key role in combating high rates of sedentary lifestyles. Teachers should be aware of promoting self-determined motivation as a way to improve students’ intentions to be physically active.	1. All variables were assessed at one point in time (cross-sectional design). Therefore, causal associations cannot be established. Longitudinal and/or experimental studies are needed to further examine the effects of the variables analyzed. 2. In order to increase knowledge about the effect of BPN in the context of PD, we suggest future studies that consider the role of need frustration on behavioral outcomes.
15. 2019 Spain	Trigueros-Ramos et al. U. Almería [35]	To analyze the teacher’s influence on confidence, enjoyment, motivation and intention to be physically active in adolescence.	The results obtained in this study support the postulates of the self-determination theory, demonstrating the predictability of the context of the PE class towards the adoption of healthy lifestyle habits, such as adequate nutrition and regular PA practice.	It would be interesting to determine the influence of the teacher’s prosocial skills on the structural dimensions, as well as their influence on the resilience and motivation of adolescents in order to know their effect on the adoption of healthy adaptive skills.	1. This is a correlational study whose results can be interpreted differently according to the reader’s understanding, so cause–effect relationships cannot be established.
16. 2019 Chile	Carrasco et al. U. Playa Ancha [36]	To establish the influence of the NPB on the physical sports practice habits of 8th grade (13 years old) elementary school students.	The data found suggest that the NPB could be factors that positively influence the promotion of physical sports practice habits in the school context.	“The proposal of a healthy physical-sports activity should be initiated from the school stage, not only as a preventive means but also as an educational-cultural phenomenon, since it has been observed that PA practiced during school hours is not enough to promote optimal health benefits”.	Does not indicate any limitation.
17. 2020 Spain	Vargas and Herrera U. Valencia [37]	To find out the motivation of adolescents towards PE and regular PA practice as a function of gender.	Boys showed greater intrinsic motivation and a higher level of habitual PA than girls. Likewise, students who habitually perform a higher level of PA have a higher level of intrinsic motivation.	The experience of positive experiences in PE together with other PA promotion strategies should be aspects to be taken into account for the reduction in sedentary lifestyles in adolescence.	1. Studies with a larger sample and in different educational contexts would be needed to corroborate these results.
18. 2020 Chile	Fuentes and Lagos U. Católica de Temuco [15]	To analyze the reasons for not practicing physical sports activities (PSA) in students between 11 and 19 years of age in the Araucanía region, Chile.	Of the 57 students who do not practice PA, 35 indicate that they stopped practicing PA due to a lack of time and interest.	It is necessary for teachers, monitors or professionals dedicated to the area of PA and health to understand the processes in which adolescents find themselves, in order to make use of their motivations and encourage the practice of PA, since young people have drastically changed their interests and these logics favor their distancing from PA practices.	1. The small number of students who were part of the sample, and it is considered that it would be a contribution to consolidate a larger number of respondents to make more robust conclusions about what motivates students not to be part of PA-related actions.
19. 2020 Estonia	Kalajas-Tilga et al. U. Tartu [38]	To examine the role of adolescents’ perception of their EF teachers’ autonomy support on adolescents’ MVPA objectively measured through motivational processes in EF.	There was a directly significant and positive relationship between autonomy support and need satisfaction. Need satisfaction positively predicted intrinsic motivation. Intrinsic motivation was positively related to MVPA. A significant indirect effect supported the mediating role of psychological need satisfaction and intrinsic motivation in the relationship between perceived autonomy support and objectively measured MVPA.	Future research in this area should include daily accelerometer measurements to provide additional information on adolescent PA. Longitudinal designs should also be used and consider others (e.g., peers, parents) as possible sources of autonomy support that may influence objectively measured AFMV.	1. The cross-sectional design of the present study, which does not allow for causal inferences between variables. Longitudinal studies are recommended to test for reciprocal effects and to determine whether the model is consistent with the results.
20. 2020 Spain	Franco and Mescardi U. Valencia [39]	To analyze adherence to sports practice, motivational climate, satisfaction of needs and behavioral regulation towards PA practice. We used a sample composed of 89 subjects, 41 boys and 48 girls aged between 14 and 18 years of high school level.	PA practice was influenced by various factors such as integrated, external and intrinsic motivation in terms of performance and gender.	They suggest orienting the PE class towards a task-oriented climate in order to increase girls’ motivation towards it. Teachers should promote a classroom climate from a globalized perspective that adapts to all types of students, avoiding closed groupings within the same group, isolation or different occupation of space based on gender or student characteristics, which would lead to significant differences in the Social Relationship.	1. Difficulty in generalizing the results, since only one educational center has been analyzed; a larger number of centers should be included.
21. 2020 China	Chen et al. U. Shanghai of Sport [40]	To test the relationship between the motivational climate of the classroom from four perspectives (autonomy support, relationship support, task-involving climate and ego-involving climate), three psychological needs.	SEM analysis revealed that task-involving climate and autonomy support were positively associated with the latter, with relatedness and competence. Relatedness support was positively related to autonomy and relatedness, whereas an ego-involving climate was only associated with competence. All three psychological needs positively affected self-determined motivation, and self-determined motivation positively affected high school students’ MVPA time in PE lessons.	It should create an EF climate that focuses on supporting autonomy (e.g., providing students with the freedom to choose), relatedness (e.g., emphasizing student cooperation), and a task-involving climate that addresses students’ effort to promote their ability, psychological needs, self-determined motivation, ultimately increasing students’ MVPA in PE.	1. The study is cross-sectional; therefore, causal inferences cannot be made. More longitudinal and intervention studies are needed. 2. Conducted only in three high schools, results cannot be used to generalize situations to other populations. Future research should increase the population. 3. The use of the SDT index to assess student motivation. Future studies should focus on analyzing the relationship between intrinsic, integrated, identified, introjected, external, amotivation and MVPA motivation in EF. 4. This study did not consider the influence of demographic variables such as gender and age when analyzing the relationship.
22. 2020 Spain	Valero-Valenzuela et al. U. Murcia [17]	To identify the motivational profiles of secondary school students and relate them to their teaching style and way of life.	The results revealed the existence of two motivational profiles in the students: “self-determined” with higher scores in intrinsic motivation and identified regulation and a “non-self-determined” profile with higher levels of demotivation, introjected regulation and external regulation. It was found that students with a more self-determined profile presented higher levels of perceived autonomy support, higher levels of intrinsic motivation and a healthier lifestyle, with a positive relationship between the autonomy support perceived by the student, their level of self-determined motivation towards sport and a healthy lifestyle.	The importance of the teacher’s performance in the classroom through autonomy support, in order to contribute to higher levels of motivation and a more active lifestyle.	1. The type of sample used was chosen for accessibility (convenience) and not in a randomized fashion, thus compromising external validity. 2. The number of participants and the study design was observational and descriptive, so that future work should consider the possibility of making similar measurements in longitudinal designs, taking into account the way the sample was selected.
23. 2020 Spain	Planas et al. Universidad Lleida [41]	Classify the timing of change in physical exercise in adolescents and analyze the relationship with motivations, barriers and physical condition.	The results show that the majority of the population studied (52.2%) were in the maintenance stage, with a difference observed in women (38.3%), and an alarming decrease in PA as age increases. The most prominent motivations are prevention and health, followed by the search for fun and wellbeing; the most evident barriers are obligations and lack of time.	Intervention programs should take these aspects into account, offering activities that are compatible with school schedules, generating motivating contexts and adapted to the stage of the subject.	1. Expand to a larger and representative sample of young people of these ages.
24. 2021 Spain	Moral-García et al. U. Pontificia de Salamanca [42]	To determine the relationships between the motivation towards PA practice of a group of adolescents in secondary education (taking into account task or ego orientation) and the variables sex, age, PA level, BMI and morphotype.	It was observed that girls and younger students are more task-oriented in PA practice, as well as obese and overweight subjects and those who consider themselves to be of the ectomorph morphotype. Boys show greater ego orientation in PA practice.	The multiple factors involved in PA practice in adolescence should be studied. It seems advisable to promote task orientation in the practice of PA in secondary education (more markedly so among boys and older adolescents), both at the school and out-of-school level, as this can lead adolescents to a higher level of practice or the maintenance of this practice in the future.	1. As they are cross-sectional in nature, it is not possible to establish causal relationships.
25. 2021 Belgium	Van Doren et al. U. Ghent [43]	To find out how students’ perceptions of the (un)motivational style of PE teachers relate to students’ device-based PA levels during the subject.	The results showed that PA was significantly predicted by individual-level factors such as self-efficacy, motivation and attitude. Among the demographic correlates, gender, age and BMI did not affect PA, whereas socioeconomic status and geographic characteristics had a significant association with PA.	It is recommended that teachers adopt a motivational style that benefits the autonomous motivation of students and classes for PE, while minimizing their controlling style to decrease the controlled motivation and demotivation of students and classes.	1. Longitudinal study to investigate causal and long-term effects. 2. Not all had accelerometers; they were given randomly. 3. Not all students provided information on their age (missing data).
26. 2021 China	Kiyani et al. U. Zhejiang /U. Rawalpindi [44]	To explore individual, interpersonal and organizational factors that may influence PA of adolescents (aged 10–14 years) in Pakistani schools.	PA was significantly predicted by individual factors such as self-efficacy, motivation and attitude. Among the demographic correlates, gender, age and BMI did not affect PA, whereas socioeconomic status and geographic characteristics had a significant association with PA.	Public health intervention strategies aimed at promoting PA in adolescents should recognize multiple levels of influences that may enhance or impede the likelihood of exercise among adolescents. In addition, parental, peer and teacher support for PA practice may provide better opportunities for adolescents to be active and reap the benefits of more meaningful PA behavior.	1. The study was cross-sectional in nature and was conducted in a single city in Pakistan, which could affect the generalizability of the results. 2. A cross-sectional design may limit causal inferences.
27. 2021 Brasil	Batista et al. U. Rio Grande do Sul [45]	To describe adolescents’ positive and negative perceptions and reasons for enjoying FE in four schools, and to identify young people’s views on possible improvements in FE classes.	Adolescents enjoy PE because of the fun, learning, enjoyment of PA, exercise and sports. Negative perceptions were related to difficulty with the teacher and not having PE classes. Adolescents suggest that they would like to have a wide variety of types of PA, physical exercise and sports modalities.	The main opinions of the adolescents on possible improvements in PE classes were to have more diversity in the classes and to increase the variety of types of movements, physical exercise and sports modalities, as well as the possibility of more dynamic classes.	1. Lack of interviews with students and teachers for a better understanding of the relationship between them. 2. Studies with complex approaches that consider psychosocial variables such as personality traits, students’ origin, body weight, satisfaction with personal appearance, and peer group relationships are needed.
28. 2021 Finland	Huhtiniemi et al. U. of Jyväskylä [46]	To investigate associations between task- and ego-involving motivational climates, perceived physical competence, physical performance, enjoyment and anxiety during two different types of EF fitness testing lessons.	The results indicated that the task-involving climate and perceived competence increased students’ enjoyment and decreased their anxiety levels, whereas the ego-involving climate had no effect on students’ enjoyment and increased their anxiety levels.	Strategies that promote motivational climate involving tasks and students’ perceived competence should be employed to increase enjoyment and decrease anxiety during EF fitness testing lessons.	1. The sample was not randomly selected. 2. The research was only applied to students in grades 5 and 8. 3. No observations or recordings of the classes were available; therefore, although the study protocol (including the order of the tests) was carefully presented to the teachers, there is no way of knowing how precisely they followed the written instructions. 4. Consider other variables such as gender, ethnicity, disability, actual or perceived level of competence, or level of physical fitness.
29. 2021 Spain	Pulido et al. U. Extremadura [47]	To analyze the association between the specific dimension (physical fitness, appearance, physical competence, physical strength and self-esteem) of students’ physical self-concept (PSC) and their PA levels (intentions to be physically active and students’ perceived levels of extracurricular PA) and sedentary behavior (SB). They also test the role of students’ motivation (intrinsic) and motivation toward PA in these relationships.	The majority of students’ PSC dimension positively predicted students’ PA scores and were negatively associated with SB. In addition, the majority of students’ PSC dimension was positively associated with intrinsic motivation and negatively predicted motivation.	Sex differences should be considered in the relationship of these variables. These differences may lead teachers to promote specific strategies to optimize the specific dimension of PSC, especially in girls, and their motivational processes with the aim of increasing their PA levels and reducing sitting time.	1. As this is a cross-sectional design, future longitudinal or experimental studies would allow for us to examine changes in the identified relationships over time and provide more solid evidence in this type of research. 2. The data in this research were collected with self-reported questionnaires by students, which may imply a social desirability bias; it would be important to analyze the association of these variables using objective measures (e.g., accelerometers).
30. 2021 Portugal	Mata et al. U. Extremadura [48]	To analyze and compare the motivational indicators of adolescents in face-to-face PE classes during COVID-19, according to gender, educational level and PA.	Differences were found in achievement goals, motivational climate and levels of motivational regulation in different groups by gender, PA and educational level, favoring older and more active participants. A more positive motivational profile was found for girls in general and, in particular, for active boys, in terms of more self-determined motivations and mastery goal orientations.	This study suggests that restrictions related to face-to-face FE classes during the COVID-19 pandemic had a negative impact on student motivation.	It is important to keep in mind that this study was primarily intended to analyze the differences between the different groups with a convenience sample, which represents an inherent limitation of the study.
31. 2022 Lithuania	Jankauskiene et al. U. Lithuanian Sports [49]	It aims to test associations between support for teacher autonomy, self-determined motivation for PE, PA habits and non-participation in PE in a sample of adolescents in a cross-sectional study.	The results show that perceived teacher autonomy support was directly positively associated with PA habits and negatively associated with non-participation in PE classes.	The practice of EF teachers in showing that supporting students’ autonomy and strengthening their self-determined motivation can facilitate greater participation in EF classes and the formation of students’ PA habits.	1. Use a larger sample of adolescents.
32. 2022 USA	Murfay et al. U. Kansas [50]	To test incoming high school students’ perceptions of the FE and how students felt former FE teachers influenced their perceptions.	Results indicate that students have a mix of positive and negative perceptions of FE and report that the purpose of the subject is to participate in PA and learn to live a healthy lifestyle through fun and meaningful experiences.	PE lessons should be designed to prioritize students who have a positive experience with PA and encourage individual competence over social comparison. Combining meaningful content with fun and successful PA experiences based on the PE standards will likely result in positive perceptions of PE and PA, and possibly increase PA behaviors in the future.	1. Interviewing students more than once and for a longer period of time could provide more information about how perceptions of the EF develop or change. 2. Moderate sample size that contained twice as many females as males.

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
