# Peer review of "Motivation toward Physical Activity and Healthy Habits of Adolescents: A Systematic Review"

_children, 2023, doi:10.3390/children10040659_

Round 1

Reviewer 1 Report (New Reviewer)

A basic point in this review article is why the authors have chosen the period of 2016 to 2022? The reason for this is not specified in the article. The research related to students' motivation to participate in physical activity has started since the self-determination theory was presented (1985), and most of the research was conducted between 2000 and 2010. Many effective results have also been obtained in this period of time, and many practical considerations have been presented to improve the level of physical activity in children and adolescents. Therefore, choosing the period from 2016 to 2022 cannot lead to finding many articles. If the time period from 2000 to 2020 was chosen, it could certainly lead to better results in this article.

The next important point is that the purpose of this review article is not clearly stated. Is the country in which the research was conducted considered a goal? Is the country of research a factor affecting the research results?

Also, health habits are mentioned in the research title, but health habits are not properly explained in the article. What is meant by health habits? Also, in the presentation of the results of the articles, health habits are not properly mentioned. In general, the variable of health habits in the present research is not clear.

In the discussion and conclusion, it is not clear why it is important in which country the research was done. In what way is the type of country important? For example, what difference does it theoretically make to the motivation of adolescents to participate in physical activity if the research was conducted in Pakistan or if it was conducted in Spain?

From a theoretical point of view, one of the most important theories related to the motivation to participate in physical activity is the self-determination theory. However, the authors have made very little reference to this theory. Self-determination theory should be fully discussed when discussing the motivation to participate in physical activity. This topic is not seen in this article.

Author Response

Reviewer 2 Report (New Reviewer)

A good review based on meta-reviews of published studies to determine which countries are leading research on adolescents' motivation towards physical activity and healthy habits and their main findings. It is essential because the development of practices during adolescence plays a significant role in how one develops as an adult. Looking at the bigger picture, it also impacts society and the country because every person would somehow influence or motivate the population around them. It is also alarming to see the increasing obesity rate in adolescent across the world, and such studies are essential to establish the usefulness of intervention and development programs looking at specific scenarios such as motivation towards healthy habits 

The authors concluded that most of the research is 19 led in Spain with 16 articles, followed by 3 in Chile, 2 in Portugal, and 2 in Norway. It would be nice to see how this research improved motivation and habits toward physical activities in adolescent populations.

A spell check is required, and it would need to be rewritten at certain places to make it easier and better for the average reader. 

Once the articles are selected from the large number of articles procured from initial searchers, what precautions were taken to ensure the inclusion and exclusion criteria were executed fairly and correctly? All these studies would be significantly different in the specific things they hypothesized. 

I understand this was a descriptive meta-review, but were there any statistical analyses done internally to see how these findings are significant?

I applaud the authors for providing their views on the recommendation and limitations of each study.

Also, what were the measures taken to see the quality of each research article that was shortlisted in the final batch that was used for the meta-review?

The discussion and conclusion parts are well defined, and I applaud the authors for having a specific section for discussing the limitations and future possibilities for the metareview. 

Author Response

Reviewer 3 Report (New Reviewer)

Children- 2295451.

Thank you for giving me the opportunity to review the manuscript entitled: « Motivation toward physical activity and healthy habits of ado-lescents: A Systematic Review »

The aim of the manuscript was to identify which countries were leading in research on adolescents' motivation towards physical activity and healthy habits. 

In general terms, I consider that the manuscript deals with an extremely interesting topic: physical activity, health and motivation in adolescent, but in my opinion it needs some revisions before publication.

Main concerns 

The manuscript is well written, easy to understand but I suggest checking for the typos, language (English) and police size (e.g., lines 56-59).

The time range delimited (selection of studies published between 2015-2022 only) choice should be justified.

I suggest to the authors to add references (please see bellow) in the introduction part of the manuscript and to evoke the countries concerned when citing previous work (e.g., Säfvenbom et al., 2012 => Norway…). This will probably provide more arguments justifying the purpose of the study.

Only two databases were examined, which could be considered a limitation since there are other relevant ones (e.g., Pubmed, Cochrane, Science direct, Google scholar…).

Another limitation to this study comes from the fact that articles published in Spanish were included, which may lead to an over-representation of publications from Spanish-speaking countries. Please consider this very important point.

All the acronyms must be defined. When using “EF” do you mean “PE” ?  (if so please use a single acronym in all manuscript for better clarity).

Suggested references 

van Sluijs EMF, Ekelund U, Crochemore-Silva I, Guthold R, Ha A, Lubans D, Oyeyemi AL, Ding D, Katzmarzyk PT. Physical activity behaviours in adolescence: current evidence and opportunities for intervention. Lancet. 2021 Jul 31;398(10298):429-442. doi: 10.1016/S0140-6736(21)01259-9. Epub 2021 Jul 21. PMID: 34302767; PMCID: PMC7612669.

Hutmacher D, Eckelt M, Bund A, Steffgen G. Does Motivation in Physical Education Have an Impact on Out-of-School Physical Activity over Time? A Longitudinal Approach. Int J Environ Res Public Health. 2020 Oct 4;17(19):7258. doi: 10.3390/ijerph17197258. PMID: 33020426; PMCID: PMC7578982.

Specific concerns

Abstract

In the abstract I suggest to evoke the time range delimited (selection of studies published between 2015-2022) for better clarity.

Introduction

Lines 33-34. A reference should be added at the end of the sentence: « These changes affect the quality of life, both in terms of healthy habits and the practice of Physical Activity (PA), in which diet and nutrition play an important role in maintaining health and preventing diseases. »  

Suggested reference

Chaudhary A, Sudzina F, Mikkelsen BE. Promoting Healthy Eating among Young People-A Review of the Evidence of the Impact of School-Based Interventions. Nutrients. 2020 Sep 22;12(9):2894. doi: 10.3390/nu12092894. PMID: 32971883; PMCID: PMC7551272

Lines 40-41. I suggest to rewrite the sentence: “Unfortunately adolescents, as indicated by the World Health Organization (WHO) [5], 1 in 4 adults and 3 in 4 adolescents (aged 11 to 17 years) worldwide are obese. » I think there is a syntax problem; I eventually suggest to remove adolescents at the biginning of the sentence. 

Lines 41-42 « As the economic development of countries increases, inactivity increases ». I also suggest to add reference.

Suggested reference

Ma C, Zhang Y, Zhao M, Bovet P, Xi B. Physical Activity and Sedentary Behavior among Young Adolescents in 68 LMICs, and Their Relationships with National Economic Development. Int J Environ Res Public Health. 2020 Oct 23;17(21):7752. doi: 10.3390/ijerph17217752. PMID: 33114035; PMCID: PMC7660305.

Line 49 « At present, no country offers the necessary conditions 49 to help all children to grow up and have a healthy future [4]. » I suggest to replace « no » by « few ». For example in Switzerland, health promotion initiatives such as "fit4future" for primary school students or "Gorilla" for young people aged 10 to 20 have been set up to combat overweight, sedentary lifestyle and provide information on balanced diet (Robin, Villoing, & Le Page, 2022).

Line 59 I suggest to remove « its » from the sentence.

Line 60 please replace « physical education » by « PE ».

Line 66 typo « thosestudents »

Material and Methods

Please give more information concerning: 

-       The Boolean phrases employed during the search. 

-       The individual who performed the searches and screened the records.

-       The search, how many authors screened the records? Did you employed interrater reliability and Cohen’s Kappe (McHugh, 2012) concerning eligibility?

-       Did the quality of the studies was evaluated? If not why?

The results part is very descriptive and a part of the results mentioned in discussion could be found in the “Result” part of the manuscript.

Conclusion

Sentences in the second and third paragraphs should not start with "that".

Round 2

Reviewer 1 Report (New Reviewer)

Dear authors,

Thank you very much for sending the revised version of your paper. Still, I have not convenienced by your response regarding including the country as an aim. In my opinion, country could not be included as a goal in a review.

Best regards,

Reviewer 3 Report (New Reviewer)

I would like to thank the authors for the changes made, which improved the quality of the manuscript. The authors responded appropriately to my suggestions and remarks.

This manuscript is a resubmission of an earlier submission. The following is a list of the peer review reports and author responses from that submission.

Round 1

Reviewer 1 Report

Title: Motivation towards physical activity and healthy habits of Chilean adolescents. Systematic Review

Article Type: Systematic Review

Summary

In this article, the authors have tried to review studies related to the factors in the incidence of motivation towards Physical Activity and healthy habits of Chilean adolescents. Following the PRISMA 2020, the authors, searched the Web of Science and Scopus databases (During September to December 2022, English and Spanish papers about these keywords: "Physical activity", "Motivation" and "Adolescents").

Evaluation

I acknowledge the authors' effort in conducting the study. However, there are some major concerns in the study that I mentioned in the below.

Please add a better results and conclusion to the abstract. At the moment, we can’t understand the results and the conclusion of the review.

Why did you not search other databases such as Medline, SportDiscus, PsycINFO?

Line 122. “The search terms used were: "Physical activity", "Motivation" and "Adolescents".”. Why did you not include some other terms like healthy habits as well as Chilean? How did you find papers about Chilean participants?

Line 131. You should add the “only Chilean participants” to the Inclusion criteria.

You wrote about 32 papers in table 1. As I understood, in first column, you have written “country”, but we can see lots of countries like Norway, etc. in the title you said that you reviewed “Chilean adolescents”, but you reviewed adolescents of other countries, why?

Author Response

Dear Evaluator, you are undoubtedly correct in your approach. As a research team we considered some of your suggestions. Within this search we wanted to identify the countries that have carried out this type of research the most and at the same time identify how much research has been done in the Chilean population in this area.

We have improved the work with your recommendations both in the title and in the general objective of the study, also the final results were deepened, being clearer with the stated objectives. But on the point raised in the search, as a research team we considered doing it with the two sources of greatest impact and relevance at the research level (JCR and SCOPUS) because normally when it is in those it is also in the others.  In relation to the search terms, only these three areas were chosen given the objectives of the study, although for future research we will consider and incorporate your recommendations and/or suggestions and incorporate other inclusion and exclusion criteria.

Reviewer 2 Report

It is suggested that the authors eliminate objective 1, since this is implicit in the systematic review, and it is suggested to restructure the conclusion, since it does not respond to what was stated in the research objectives, which would be 2 if the objective is eliminated one .

What are the main recommendations of the authors?

What are the main limitations found by the authors?

The rest, in my opinion, is correctly structured.

Sincerely

Author Response

Dear reviewer, in relation to your suggestions we believe that your contributions will help us to improve the article, both in the title and in the general objective of the study. Also the final results were deepened, being coherent with the stated objectives, elements that will improve the analysis of the conclusions incorporating more specific aspects in relation to the other objectives such as recommendations and limitations. And finally, in your suggestion to eliminate the first objective, we made an adaptation given that it would lose the central sense of the search, considering that it should be explicitly stated in order to be able to respond to it in the discussion and conclusions, although we modified it to make it more coherent with our purposes.

Reviewer 3 Report

The manuscript entitled " Motivation towards physical activity and healthy habits of Chilean adolescents. Systematic Review " is a great review on the topic of motivation for physical activity. I believe it could add significant information to the existing literature. However, I recommend changing the title. The Authors would like to investigate Chilean adolescents, but only 3 studies were found, so they analyzed other studies as well. This makes the study important, but this review shows more than Chilean adolescents.  The introduction is well written, but please introduce the physical education system in Chile in this part. Please use other words than incidence (line 105) since it is mostly used in epidemiology studies. The methods are clear and easy to understand how the Authors got their results. Although the results contain only one table. I recommend highlighting the most important part of the table. The discussion is following scientific standards and the Authors use several kinds of literature to prove their points. I have only one recommendation for this part. Add practical suggestions to practitioners according to your findings.  

Author Response

Dear Reviewer, we are grateful for your recommendations, which will undoubtedly help us to improve our publication by making the recommendations indicated as you can see. Within this search, given that only 3 articles referring to Chilean adolescents appeared, we wanted to identify the other countries that have carried out this type of research. Their recommendations have improved both the title and the general objective of the study, also the final results were deepened, remaining consistent with the objectives set out in the final conclusions. where we also incorporate practical suggestions. In relation to highlighting the table can be seen in the scope of the results highlighted in the table. Regarding the suggestion to incorporate aspects of PE in Chile in the introduction, it was incorporated in the introduction.

Round 2

Reviewer 1 Report

I acknowledge the authors effort for revision, however, I think the authors did not resolve the concerns, so I think this manuscript cannot be accepted for publication.

Author Response

Dear reviewer, we would like to thank you for your comments to improve the English language, when making the modifications of the (previous) reviewers, these were not translated correctly when linking them in the paragraphs. We would like to make your suggestions, which can be seen in the document.